# Validating aggregative soil sampling using bootie and drag swabs hydrated with simple wetting agents in commercial produce fields

Erin Kealey,[1] Ray Elementi,[1] Zemme Frankowski,[1] Negin Valizadegan,[2] Cecil Barnett-Neefs,[1] Jiaying Wu,[1] Pratik Banerjee,[1] Matthew J. Stasiewicz[1]

**ABSTRACT**  Bootie and drag swabs may collect more microbiologically representative aggregative soil samples than composite grabs from produce fields. Previous experimental field work has identified some practical wetting agents as potential alternatives to traditionally used skim milk. This study validates the two most promising wetting agents, phosphate-buffered saline (PBS) and buffered peptone water (BPW), by comparing swabs results from 100 m tracks through a melon farm (262,000 $m^2$), a mixed agriculture farm (leafy green, peppers, beets, 4,500 $m^2$), and an apple orchard (114,000 $m^2$). The mean difference between paired samples collected with BPW or PBS ranged from −0.02 ± 0.09 to 0.26 ± 0.09 $\log_{10}(CFU/g)$ for aerobic plate count (APC) and total coliforms (TCs). Bootie and drag swabs recovered greater APCs [mean difference 0.63 ± 0.38 to 1.83 ± 0.24 $\log_{10}(CFU/g)$ and TCs (mean difference 1.32 ± 0.96 to 5.32 ± 1.03 $\log_{10}(CFU/g)$], and greater prevalence of *Escherichia coli* compared to soil grabs (90% versus 44% of samples positive by enrichment, $P < 0.001$). By 16S sequencing, samples collected with PBS had greater within-sample community richness (alpha diversity) than BPW (*P*-values 0.041 and 0.059) but similarly overlapping taxa. Soil samples had higher within-sample (alpha) diversity ($P < 0.05$), but lower between-sample (beta) diversity compared to booties and drags. Overall, there was no biologically meaningful difference between the performances of the two wetting agents for bootie and drag swabs, and compared to composite soil samples, these two swab methods recovered more indicator organisms from produce field soil representing five different commodities.

**IMPORTANCE**  Bootie and drag swabs have emerged as a promising alternative to composite soil grabs for produce industry use for improved preharvest soil sampling for safety and quality purposes. This study builds on previous work in research field trials to identify practical wetting agents for the bootie and drag swabs because the skim milk powder classically used in animal operations is not practical for produce. Here, we validate the two most promising, practical wetting agents previously identified (phosphate-buffered saline and buffered peptone water), testing their use in commercial settings, sampling soil from melon, leafy green, beet, pepper, and apple production systems. By showing the wetting agents perform similarly, and that booties and drag swabs are at least as good or better than composite grabs at recovering indicator organisms, we have identified a viable method for improved agricultural soil sampling for microbiological profiling.

**KEYWORDS**    environmental sampling, produce safety, safety indicator, quality indicator, soil, aggregative sampling, food safety

Address correspondence to Matthew J. Stasiewicz, mstasie@illinois.edu.

The authors declare no conflicts of interest. The funders had a role in gaining access to melon farms for sample collection but no role in the design of the study, in the collection, analyses, or interpretation of data, in the writing of the manuscript, or in the decision to publish the results.

See the funding table on p. 14.

Improving preharvest soil sampling in produce fields by making it more powerful, efficient, and practical could encourage additional food safety and quality monitoring, allowing for potential corrective actions to reduce contamination before the produce is harvested. There are many routes of contamination of produce from soil (1–3). The consumption of raw produce, such as row crops, fruits, and vegetables, has been associated with several foodborne illness outbreaks, and the rate of illness has been steadily increasing over time (4). Therefore, monitoring soil before harvesting produce could provide another route for data collection for indicator organisms, thus improving food safety and quality in the produce industry.

Indicator organisms are used in microbiological testing research programs in industry, as they may signify the potential presence of pathogens in a sample (5). In soil, indicator organisms commonly used to assess quality and safety include aerobic plate count (APC), total coliforms (TC), and *Escherichia coli*. APC is a quality indicator and intended to indicate the general microbial load in a sample (5). TC and *E. coli* are safety indicators, and their presence could indicate the presence of fecal contamination or potential pathogens in a sample (6, 7). Understanding the specific microbial communities in soil can aid in the development of biologically relevant monitoring tools that support produce growers by bringing specificity to potential environmental issues associated with soil, including detecting the presence of microorganisms related to human consumption and health (8).

In produce fields, soil sampling is conducted for several reasons, including collecting soil cores to analyze physical and chemical properties (9). Collecting these cores is labor-intensive and only captures a very small portion of the entire field (10); therefore, aggregative samples could increase the area of a field sampled. Currently, aggregative sampling is regularly used in the meat and poultry industry to monitor for the presence of pathogens, or their indicators, in large-scale production without destroying the product. For example, bootie swabs hydrated in skim milk are the standard in the poultry industry for detecting fecal indicators in poultry litter and evaluating cleaning practices in poultry houses (11). Bootie swabs are made of cotton with an elastic attached around the top and can be worn over boots to collect surface-level samples while the sampler walks (10, 12, 13). Recent research shows aggregative bootie cover sampling could be an important tool in sampling soil for preharvest produce safety, where aggregative sampling methods, such as bootie swabs, perform better than high-resolution soil samples in active lettuce fields, even when not hydrated (10). Building upon the previous study, reference 13 evaluated the efficacy of five different wetting agents (phosphate buffered saline, buffered peptone water, skim milk, sterile deionized water, and tryptic soy broth; to find an alternative to skim milk) for bootie and drag swabs. That study found there was not a biologically meaningful difference [$<1 \log_{10}$ (CFU/g)] between the wetting agents' ability to detect indicator organisms (APC, TC, and *E. coli*) and that hydrated aggregative bootie covers performed better than the soil grabs overall. Our study utilized methodology from these previous works and used them directly in commercial produce fields to understand their functionality within an industry context. This study seeks to validate aggregative bootie and drag swab sampling in commercial-scale produce fields.

Phosphate-buffered saline (PBS) and buffered peptone water (BPW) were the wetting agents selected for validation for their osmotic balancing properties (14) and non-selective nutritive properties (15), respectively. The objective of this study is to compare and validate the use of PBS or BPW as wetting agents for aggregative bootie and drag swabs for detecting three different indicator organisms in a variety of commercial-scale produce fields against surface-level soil grabs. The sample collection method effectiveness of detecting indicator organisms for each wetting agent was also evaluated across the different commercial fields.

## RESULTS

Overall, bootie and drag swabs recovered similar, or higher, levels of indicator organisms for quality (APC) and safety (TC) than soil grabs, regardless of the wetting agent (BPW or

PBS) used to hydrate the swabs (Fig. 1). In addition, *E. coli* levels were much lower than TC and APC levels.

## Wetting agents performed similarly for detecting aerobic plate counts and total coliforms

Figure 2 shows the relationship between the $\log_{10}$(CFU/g) measured between paired samples separated by test type (TC or APC). The APC linear regression $R^2$ of 0.37 and the Pearson's correlation coefficient of 0.61 mean there is a weak, positive correlation for APC between paired BPW and PBS. The TC linear regression $R^2$ of 0.72 and Pearson's correlation coefficient of 0.85 mean there is a positive, stronger, linear relationship between paired samples. Overall, TC had a stronger correlation than APC between paired samples collected with two different wetting agents (16). Paired comparison *t*-test analysis checked for an overall difference in performance between PBS or BPW (Table 1). Generally, there was no statistically significant difference ($P > 0.05$) in mean for APC or TC between the wetting agents for drag or bootie swabs when comparing paired sample types (i.e., comparing PBS drags to BPW drags for APC), ranging from −0.02 to 0.14 $\log_{10}$(CFU/g) for mean difference of paired samples for APC booties, APC drags, and TC booties. One exception was TC results for drag swabs, where BPW collected, on average, 0.26 $\log_{10}$(CFU/g) (FDR corrected $P = 0.02$) more TC than the swabs hydrated with PBS.

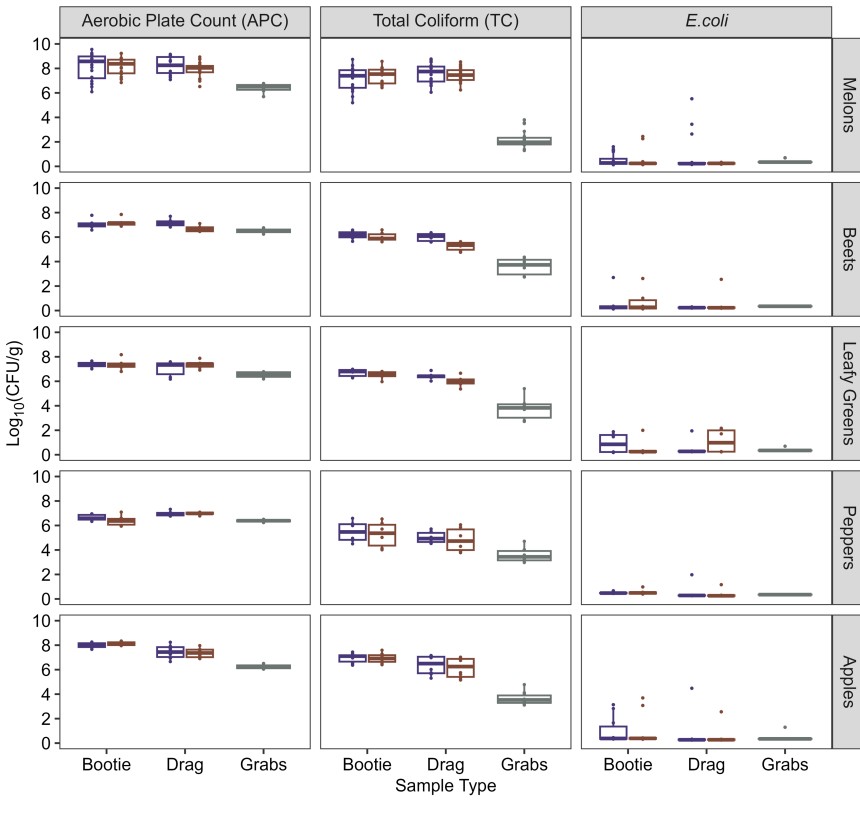

**FIG 1** Summary of enumeration results for each sample type, wetting agent, and commodity. Soil collected using three different sample types (bootie, drag, and grabs) was enumerated [$\log_{10}$(CFU/g)] for aerobic plate count (APC), total coliforms (TC), and *E. coli*. Bootie and drag samples were hydrated with either BPW or PBS as wetting agents, and soil grabs were not hydrated (color). Melons, beets, leafy greens, peppers, and apples were the produce commodities that were growing on the fields where the samples were collected. The horizontal line inside the boxes represents the mean; boxes represent interquartile range; and whiskers show 1.5× the interquartile range. If *E. coli* data points were measured below the LOD of ~0.7 $\log_{10}$(CFU/g), results for that sample were plotted and analyzed at 1/2 LOD of ~0.35 $\log_{10}$(CFU/g).

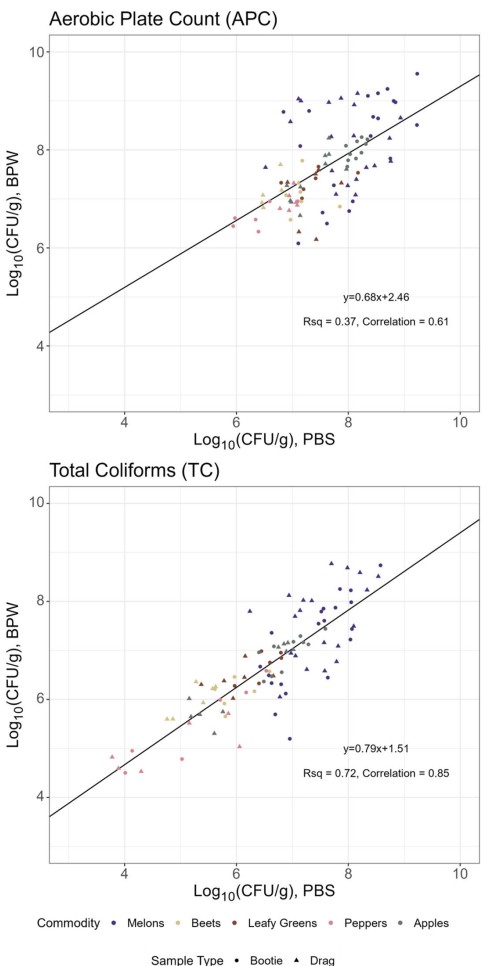

**FIG 2** Comparing the performance of PBS and BPW wetting agents for recovering APC and TC. Plotted points are matched pairs of enumeration results (log$_{10}$[CFU/g]) of aerobic plate counts (APC; top) and total coliforms (TC; bottom) for each sample type, drag or bootie swabs (point shapes) taken with each wetting agent, and PBS and BPW (axes) from different commodity soils (point colors). The black line is the linear regression, with slope, $R^2$, and Pearson's correlation coefficients annotated. Matched-pair comparison analyses are in Table 1.

However, even the largest mean difference of 0.26 log$_{10}$(CFU/g) is not likely biologically meaningfully different [using a threshold of 1 log$_{10}$(CFU/g)] to indicate a biologically meaningful difference.

Quantitative relationships were supported by trends in microbial diversity within samples, specifically the alpha diversity metrics of observed and Chao1. The mean community richness from samples collected with PBS was slightly greater than those collected with BPW, with *P*-values for observed and Chao1 being 0.059 and 0.041, respectively (Fig. 3, top), with a large overlap in the range of sample diversity and species richness. The observed alpha diversity metric shows community richness, and Chao1 demonstrates richness while also accounting for rare taxa that are present. This means that most observed taxa detected when using PBS or BPW as wetting agents are not significantly different and are only significantly different when rare occurrences are considered. However, while a PCoA Bray-Curtis distance shows a statistically significant compositional difference between sample types (permutational multivariate analysis of variance [PERMANOVA], *P* = 0.004, Fig. 3, bottom), it does not appear that there is a meaningful difference in composition between the samples collected with different wetting agents (large overlap between regions and sample points).

**TABLE 1** Paired comparison analysis for soil samples collected with BPW or PBS as wetting agents for each sample collection type[a]

| Test type | Sample collection type | Mean difference of paired samples, (BPW − PBS) $\log_{10}$ (CFU/g) ± SE | *P*-value | False discovery rate (FDR) corrected *P*-value |
|---|---|---|---|---|
| APC | Drag | 0.14 ± 0.11 | 0.197 | 0.369 |
| | Bootie | −0.02 ± 0.09 | 0.812 | 0.818 |
| TC | Drag | 0.26 ± 0.09 | 0.005 | 0.020 |
| | Bootie | −0.02 ± 0.07 | 0.818 | 0.818 |

[a]Source data plotted in Fig. 2.

## Drag and bootie swabs are more effective than soil grabs at detecting aerobic plate counts and total coliforms

Overall APC ranged from 6.52 ± 0.07 llto 8.15 ± 0.11 $\log_{10}$(CFU/g) for booties, 6.93 ± 0.10 to 8.04 ± 0.11 $\log_{10}$(CFU/g) for drags, and 6.24 ± 0.10 to 6.53 ± 0.17 $\log_{10}$(CFU/g) for soil grabs. Overall TC ranged from 5.38 ± 0.23 to 7.27 ± 0.12 $\log_{10}$(CFU/g) for booties, 4.94 ± 0.23 to 7.54 ± 0.12 $\log_{10}$(CFU/g) for drags, and 2.21 ± 0.17 to 3.97 ± 0.22 $\log_{10}$(CFU/g) for soil grabs (Table 2). Generally, bootie and drag swabs detected similar concentrations of indicator organisms for each commodity type, as there was no significant difference (similar letters in Tukey's HSD comparison) between the average $\log_{10}$(CFU/g) of APC or TC detected by drags and booties for melons, beets, leafy greens, and peppers (four out of five commodities). Booties detected slightly higher levels than drags of APC and TC for apples. In addition, for two of the five commodities, soil grabs detected lower average APCs than booties or drags, and average TCs from soil grabs were lower than booties and drags for all five commodities.

For all but one specific comparison, bootie and drag swabs recovered significantly higher levels of APC or TC than soil grabs (Table 3, all FDR corrected *P*-values from *t*-tests were <0.05 with *P*-values listed in Table S1). The one exception was for bootie swabs detecting no significant difference compared to soil grabs for APC in peppers [difference of 0.13 ± 0.39 $\log_{10}$(CFU/g), *P* = 0.24; Table 3]. APC differences between booties and soil ranged from 0.13 ± 0.39 to 1.83 ± 0.24 $\log_{10}$(CFU/g), and the difference between drags and soil ranged from 0.42 ± 0.52 to 1.63 ± 0.74 $\log_{10}$(CFU/g). TC differences between booties and soil range from 1.76 ± 1.05 to 5.05 ± 1.06 $\log_{10}$(CFU/g), and the differences between drags and soil ranged from 1.32 ± 0.96 to 5.32 ± 1.03 $\log_{10}$(CFU/g). In summary, for each of the commodity types, bootie and drag swabs recovered similar or greater levels of APC and TC compared to the soil grabs. Comparing results between commodities, soil from melons consistently showed the largest difference in recovery of APC and TC for bootie or drags compared to grabs (Table 3, noting Tukey's HSD results as capital letters), and in the bootie-grab APC comparison, results from soil from apples were not significantly different from melons. For the remaining four commodities, there were more similarities between mean differences of detected indicator organisms; however, soil from apples generally had the next greatest differences detected, then followed by leafy greens, beets, and peppers (additional information in Table S2).

Considering microbial diversity within and between sample collection methods (Fig. 4), soil grabs were more diverse than bootie and drag swabs (*P* < 0.05) for observed and Chao1 alpha diversity metrics (top 10 taxa plotted in Fig. S1). Bootie and drag swabs had similar community richness; however, there was a trend toward the mean values of alpha diversity of the booties being slightly greater than those of drag swabs, as alpha diversity *P*-values for observed and Chao1 are 0.035 to 0.039, respectively (Fig. 4, top). For beta diversity, the PCoA plot of the Bray-Curtis distance (Fig. 4, bottom) shows that soil grabs largely clustered very similar to each other, while bootie and drag swabs had a similar and much wider range of variability. Considering both microbial diversity and indicator organism levels, soil grabs detect greater microbial diversity than the bootie

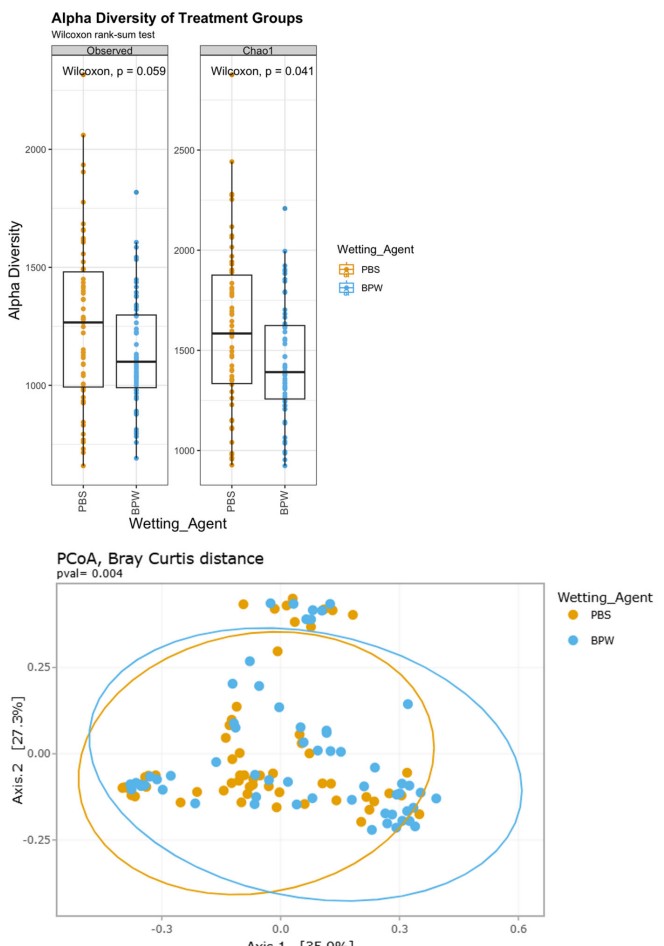

**FIG 3** Alpha and beta diversities of 16S V3–V4 sequences comparing swab data by wetting agent. (Top) Alpha diversity metrics, observed and Chao1, and Wilcoxon signed-rank comparisons between samples with different wetting agents, PBS or BPW. Each box represents the interquartile range; the horizontal line inside the box is the mean; whiskers show 1.5× the interquartile range; and each dot represents an individual sample. (Bottom) Beta diversity analyses of microbiome compositional differences between the soils collected using swabs hydrated with different wetting agents, shown as PCoA Bray Curtis distance. The ellipses show a 95% CI.

and drag swabs; however, the aggregative bootie and drag swabs were more sensitive to the target produce safety and quality indicator organisms (APC, TC, and *E. coli)*.

## Enrichment results show that bootie and drag swabs test positive significantly more frequently for *E. coli* than soil grabs

There was a statistically significant difference ($x^2$ $P = 2.9 \times 10^{-13}$) in *E. coli* positives between the combined bootie and drag samples (173 positive out of 192 samples) compared to soil grabs (21 positives out of 48 samples), meaning that for these data (Table 4), it was more likely for soil grab samples to test negative for *E. coli* than hydrated swab methods. This means that the bootie or drag swab methods are more effective at detecting fecal indicators, and samplers should consider using these methods if they wish to monitor for the presence of these microorganisms.

**TABLE 2** Indicator organism results by agricultural commodity soil type and collection method[a]

| Soil from given commodity | APC ± SD, log$_{10}$ (CFU/g) | | | TC ± SD, log$_{10}$ (CFU/g) | | |
|---|---|---|---|---|---|---|
| | Bootie | Drag | Grab | Bootie | Drag | Grab |
| Melons | 8.15 ± 0.11 a | 8.04 ± 0.11 a | 6.41 ± 0.16 b | 7.27 ± 0.12 a | 7.54 ± 0.12 a | 2.21 ± 0.17 b |
| Beets | 7.13 ± 0.10 a | 6.93 ± 0.10 a | 6.51 ± 0.14 a | 6.09 ± 0.14 a | 5.62 ± 0.14 a | 3.59 ± 0.20 b |
| Leafy greens | 7.36 ± 0.12 a | 7.21 ± 0.12 a | 6.53 ± 0.17 a | 6.60 ± 0.16 a | 6.21 ± 0.16 a | 3.97 ± 0.22 b |
| Peppers | 6.52 ± 0.07 a | 6.97 ± 0.07 a | 6.38 ± 0.10 a | 5.38 ± 0.23 a | 4.94 ± 0.23 a | 3.61 ± 0.33 b |
| Apples | 8.07 ± 0.07 a | 7.40 ± 0.07 b | 6.24 ± 0.10 c | 6.94 ± 0.10 a | 6.26 ± 0.13 b | 3.64 ± 0.19 c |

[a]ANOVA to compare the mean APC or TC by each sample collection method (bootie, drag, or grab) for each commodity (melons, beets, leafy greens, peppers, apples). Means for the same test within the same row not sharing the same subscript letter indicate values are significantly different ($P < 0.05$) by *post-hoc* Tukey's HSD test.

## DISCUSSION

### Wetting agent selection can be based on convenience and practicality

This study builds off of previous studies that evaluate the possibility to replace soil grabs with hydrated bootie and drag swabs with different wetting agents (skim milk, tryptic soy broth, BPW, PBS, and deionized water) on sampling fields fertilized with manure to ensure high indicator organism counts (10, 13). Our study aimed to validate the use of both PBS and BPW by sampling in commercial produce fields. Broadly, it was found that PBS and BPW performed very similarly across the five different commodities and that there were no biologically meaningful mean differences [<1 log$_{10}$(CFU/g) differences] as wetting agents. Quality and safety indicator counts detected by the bootie and drag swabs were greater than those detected by aggregative soil grabs, and microbial community analysis shows high similarity in the composition and diversity of taxa detected in bootie and drag swab samples. Other studies compare the use of preharvest soil sampling for food safety purposes; however, each study had a specific target organism or purpose in mind other than general indicator organism detection, thereby using wetting agents that have specific nutritive or selective properties (17–19). For example, one study utilized drag swabs hydrated in full-strength evaporated skim milk in almond orchards; however, the main goal of this study was to detect *Salmonella* Enteritidis phages specifically (18). The benefit of using PBS or BPW is that they are both non-selective, osmotically balanced, and non-allergenic, and are, therefore, more practical than the other commonly used wetting agents of skim milk, which is allergenic and animal-based, thus introducing potential undesirable properties for some produce growers (13). Therefore, either PBS or BPW could be used for the wetting agent to collect aggregative soil samples with bootie or drag swabs, and either will detect similar levels of indicator organisms. Thus, a wetting agent can be selected for preharvest soil sampling based off convenience or practicality.

**TABLE 3** Differences between the aggregative sample methods and soil grabs for indicator organisms by commodity

| Soil from given commodity | APC ± SD, log$_{10}$ (CFU/g) | | TC ± SD, log$_{10}$ (CFU/g) | |
|---|---|---|---|---|
| | Bootie – grab[a] | Drag – grab[a] | Bootie – grab[a] | Drag – grab[a] |
| Melons | 1.74 ± 0.94 A | 1.63 ± 0.74 A | 5.05 ± 1.06 A | 5.32 ± 1.03 A |
| Beets | 0.62 ± 0.38 B | 0.42 ± 0.52 C | 2.50 ± 0.88 BC | 2.03 ± 0.91 BC |
| Leafy greens | 0.83 ± 0.37 B | 0.68 ± 0.48 BC | 2.80 ± 0.93 BC | 2.42 ± 1.03 BC |
| Peppers | 0.13 ± 0.39[b] B | 0.59 ± 0.17 BC | 1.76 ± 1.05 C | 1.32 ± 0.96 C |
| Apples | 1.83 ± 0.24 A | 1.16 ± 0.44 B | 3.30 ± 0.65 B | 2.62 ± 0.93 B |

[a]One-way ANOVA to test (by column) if there was a significant difference between commodities for the difference between bootie – grab or drag – grab for APC or TC. Means within the same column not sharing the same subscript letter indicate values are significantly different ($P < 0.05$) by *post-hoc* Tukey's HSD test.
[b]Paired *t*-tests were conducted to test if the mean differences between bootie swabs and soil grabs or drag swabs and soil grabs were statistically different from 0. The difference for APC from peppers showed that there was no significant difference ($P = 0.24$) between bootie swabs and soil grabs. All other mean differences were significantly different from 0 (FDR $P < 0.05$ for each value).

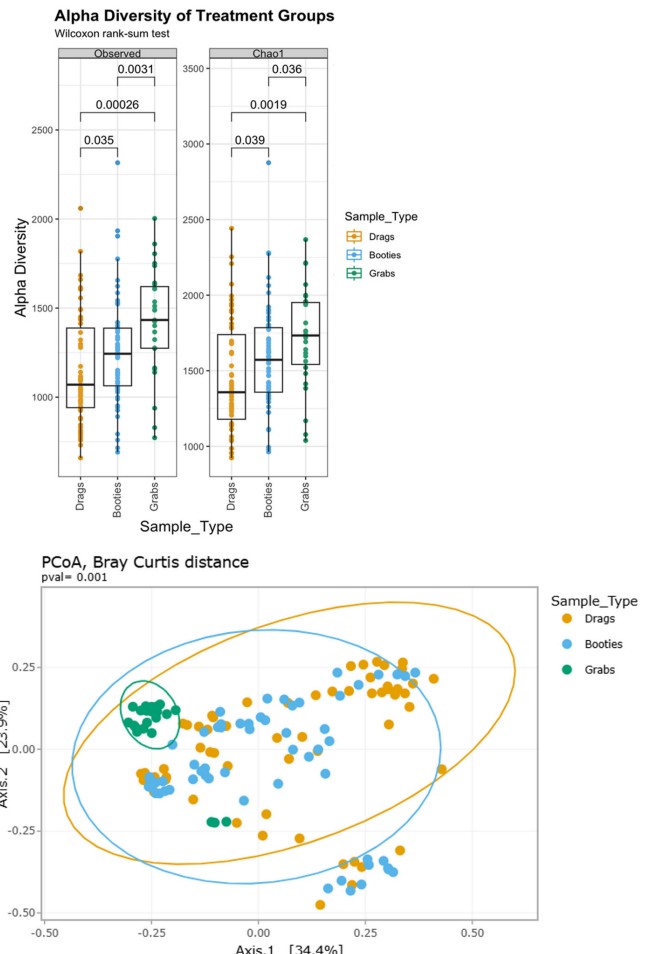

**FIG 4** Alpha and beta diversities of V3–V4 targets by sample type. (Top) Alpha diversity metrics, observed and Chao1, and Wilcoxon signed-rank comparisons between samples with different sample types. Each box represents the interquartile range; the horizontal line inside the box is the mean; whiskers show 1.5× the interquartile range; and each dot represents an individual sample. (Bottom) Beta diversity analyses of microbiome compositional differences between the soils collected through the sample collection method, shown as PCoA Bray Curtis distance. The ellipses show a 95% CI.

## Aggregative sampling methods outperform grab sampling in produce field soil and other food production systems

The use of aggregative soil sampling from bootie and drag swabs in commercial produce field was inspired by non-destructive sampling methods utilized in the beef and poultry industry. Results show that bootie and drag swabs in the produce field setting are more sensitive and detect higher levels of APC, TC, and *E. coli* in comparison to soil grabs, further reinforcing that the swab type sampling is more robust than traditional soil grabs, similar to concepts observed in the poultry and beef industries. For example, drag swabs are based off of continuous FSIS non-destructive beef trim sampling and manual sampling devices (20). Reference 21 found that continuous sampling devices were as effective or better than manual sampling of beef trim likely due to the increased surface area sampled. Bootie swabs are effective for collecting aggregative samples across multiple dairy herds or pens when monitoring for *E. coli* (12). Bootie swabs hydrated in skim milk typically are used in poultry barns (also known as poultry houses) to detect fecal indicators to evaluate hygiene management. Using bootie swabs, samplers are more efficient and accurate, as traditional sponge-swab sampling in poultry barns requires more effort to sample a small area of the floor (11). These animal models' results

**TABLE 4** Contingency table comparing presence of *E. coli* enrichment results (positive or negative by count) by wetting agent and collection type by wetting agent

| Wetting agent | Collection type | Positive *E. coli* | Negative *E. coli* | Total |
|---|---|---|---|---|
| PBS | Drag swab | 45 | 3 | 48 |
| | Bootie swab | 42 | 6 | 48 |
| BPW | Drag swab | 42 | 6 | 48 |
| | Bootie swab | 44 | 4 | 48 |
| **Swab total** | | **173**[a] | **19**[a] | **192** |
| **Soil only** | Grabs | **21**[a] | **27**[a] | **48** |
| **Total** | | **194** | **46** | **240** |

[a]Chi-square test showed a significant difference ($P = 2.9 \times 10^{-13}$) between positive and negative *E.coli* counts for swab total and soil grabs. The numbers with superscripts are the values used for the statistical test. The bolded cells are the fully annotated contingency table used for this statistical test.

are consistent with our finding that the use of aggregative soil sampling with hydrated bootie or drag swabs better samples a large area of commercial produce fields.

While microbial community analysis shows that soil grabs were more diverse, bootie and drags performed better by detecting higher levels of the target indicator organisms APC, TC, and *E. coli* by enumeration and enrichment. This could be because the bootie and drag swabs sample only the surface of the soil, which is the closest point of contact to the surface of the produce. In contrast, soil grabs are sampled at a depth up to 5 cm and, therefore, may have sampled a greater variety of taxa found under the surface (22). As detecting safety and quality indicator organisms was the primary focus of this study, aggregative bootie and drag swabs are the more sensitive and preferred sample collection method in this case, as they collected statistically significantly or biologically meaningfully higher levels of the quality and safety organisms.

For functionality purposes, collecting soil samples using the swab method has additional benefits. One benefit is not needing to bend over to manually collect a surface sample of soil multiple times across the field; instead, samplers could use the bootie and drag swabs to collect samples as they walk. Our team observed that the drags and booties were easier and more practical than the soil grabs when following this protocol on a commercial scale. Additionally, aggregative sample collection using bootie swabs may present less of a trip hazard, as the drag swabs tend to get caught on plants or debris and would occasionally get tangled behind the sampler as they walked. As bootie and drag swabs gave similar microbiological results, swab type can also be chosen for preference or convenience. A limitation of this study is that the methodology of this experiment only enumerated the drag and bootie swabs independently of each other, and the option of a combined bootie and drag swab was not evaluated.

## Aggregative soil sampling is useful across multiple commodity types

Five commodities were sampled for this study: leafy greens, peppers, melons, apples, and beets. Each commodity has a different ground preparation, which may present different challenges to the swabs in recovery of indicator organisms. Therefore, evaluating the effectiveness of bootie swabs, drag swabs, and soil grabs for each of these produce types is important to understand the versatility of these methods in terms of detecting different safety and quality indicator organisms.

Leafy greens were available on the small, mixed-horticulture field, and samplers were able to collect soil near the produce for this commodity. In addition, the sampling protocol used in this study was originally used in a lettuce field (10) to compare the functionality of bootie swabs and conventional soil grabs. As the edible portion of the leafy greens is in contact with the soil, there is a risk of transmitting pathogens from the soil to the product surface. In addition, besides direct contact, additional contamination can come from splashing of soil to the surface due to rainfall. Peppers were also grown at this location, and as they grow in rows and are close to the ground, they were sampled in the same manner as the leafy greens. Bennett et al. (4) state that between the years 1997

and 2013, solanaceous vegetables, such as peppers or tomatoes, were responsible for 47 foodborne disease outbreaks, and, therefore, the peppers were also reasonable to study.

Fruits, such as melons, that have inedible skin and grow on top of the soil's surface have caused foodborne disease outbreaks. According to the Food and Agriculture Organization (FAO) (23), 85 outbreaks of foodborne illness were identified linked to melons from 1950 to 2011. The FAO stated that melons have specific characteristics on their rinds, such as a highly textured topography from the netted surface or highly waxy and hydrophobic surface, which can prevent microorganisms from being washed off with water or being protected from post-harvest sanitizers. Once these melons are harvested, the slicing of the fruit can expose the inner, edible flesh from the contamination from the rind (23).

Fruiting trees can be evaluated under a different lens, as the edible portion of the plant is higher from the surface of the soil. In addition, soil sampling for fruiting trees could be an indicator of soil health in general, in addition to safety monitoring (8). Grounded fruit, or fruit that has come in contact with the soil, cannot be harvested by growers unless there are further processing steps according to the Produce Food Safety Rule (PSR) of FSMA (24). However, in some cases, short or small apple trees that have branches containing fruit can be grown very close to the ground without proper pruning. The PSR clearly covers guidelines for dropped produce, but it does not mention protocols for drooping produce that is still attached to the parent plant but is either on or very close to the ground. Drooping produce can still increase the risk of the presence of biological presence on the surface of the skin (24). Other contamination routes for fruiting trees may come from harvesting by hand (25), especially if sampler hands are not washed properly or alcohol-based hand sanitizers are not used (26).

Beets were grown on the small, mixed-horticulture field where leafy greens and peppers were also sampled, and, therefore, the samplers decided to follow the protocol in this area to see if there was a difference in detected indicator organisms in comparison to the other commodities. While beets are on the "rarely consumed raw" list and are not subject to CFR § 112.2(a)(1) of the FSMA PSR (27), they could be considered for preharvest soil monitoring. Other produce items, such as carrots or onions, grow in a similar manner to beets but are regularly eaten raw. Onions have been linked to foodborne disease outbreaks, such as the raw sliced onions contaminated with *E. coli* O157:H7 (28). Therefore, our bootie or drag swab sample methods for root vegetables that are eaten without additional cooking could still apply. For other commodities that are grown but still occasionally eaten raw, studies by Bennett et al. (4) compare the frequency of foodborne illness outbreaks due to root vegetables in comparison to other produce categories. They found that while the proportion of foodborne illness outbreaks due to the consumption of raw root vegetables, such as beets, is low in comparison to the total number of outbreaks from produce from the years 1998–2023, it is still important to consider that foodborne illness is possible from beets and other commodities that grow in the ground.

## MATERIALS AND METHODS

### Swab construction and preparation

Three sample collection methods were evaluated during this study: cotton bootie covers, drag swabs, and aggregate soil grabs. Sterile cotton bootie covers were sourced commercially (CAT#10001905, Romer Labs, Union, MO). Drag swabs were constructed from a 4 × 4 in. sterile gauze pad and 1.2 m of autoclaved cotton kitchen twine tied to the corner, as described by Wu et al. (13), with the assembled swab placed in a clear plastic bag (CAT# B01020, Whirlpack, Uline, Pleasant Prairie, WI), and then disinfected under UV light for 15 min (29). Soil grabs were collected with a clean, disinfected handheld metal spade (10, 13).

## Sample collection method

Each sampling repetition consisted of a matched set of one bootie swab and one drag hydrated with sterile phosphate-buffered saline (PBS) (Sigma-Aldrich, St. Louis, MO) worn by the sampler on their right leg, one boot and one drag swab hydrated with sterile buffered peptone water (BPW) (Millipore, Burlington, MA) on their left leg (Fig. 5), and one composite soil sample. Each bootie and drag were pre-hydrated with 18 or 12 mL of the designated wetting agent, respectively, prior to arriving at the sampling field. Plastic boot covers (CAT#10001917, Romer Labs, Union, MO) worn under the cotton bootie swabs were changed after each repetition to prevent cross-contamination from the bottom of the samplers' shoes. Disposable, nitrile gloves were changed and disinfected with 70% ethanol before collecting each repetition of samples (30). Composite soil grabs consisted of six soil grabs taken by scooping approximately 50 g of surface soil, no deeper than 5 cm, six times during the walk of the path using a handheld spade that is cleaned and disinfected with 70% ethanol between each repetition. After samples were collected, they were placed in a cooler with ice packs and kept at refrigerated temperatures (~4°C) until processing. A brief study was conducted to evaluate the stability of APC, TC, and *E. coli* after sampling but before processing, as previous methodology states that samples must be processed within 24 h (13). Twelve samples were held at 4°C for 7 days and tested at days 0, 1, 2, and 7 using the protocols described below. It was found that the samples were stable up to day 2 or 48 h after collection (for additional details, see Fig. S2).

## Sample collection walking patterns

For melons, 20 repetitions were collected over a 2-day sampling trip to a commercial melon farm in the United States (10 repetitions per day) in May 2024. The sample field was approximately 750 m wide and 300 m long (Fig. S3). Every fourth bed was used as

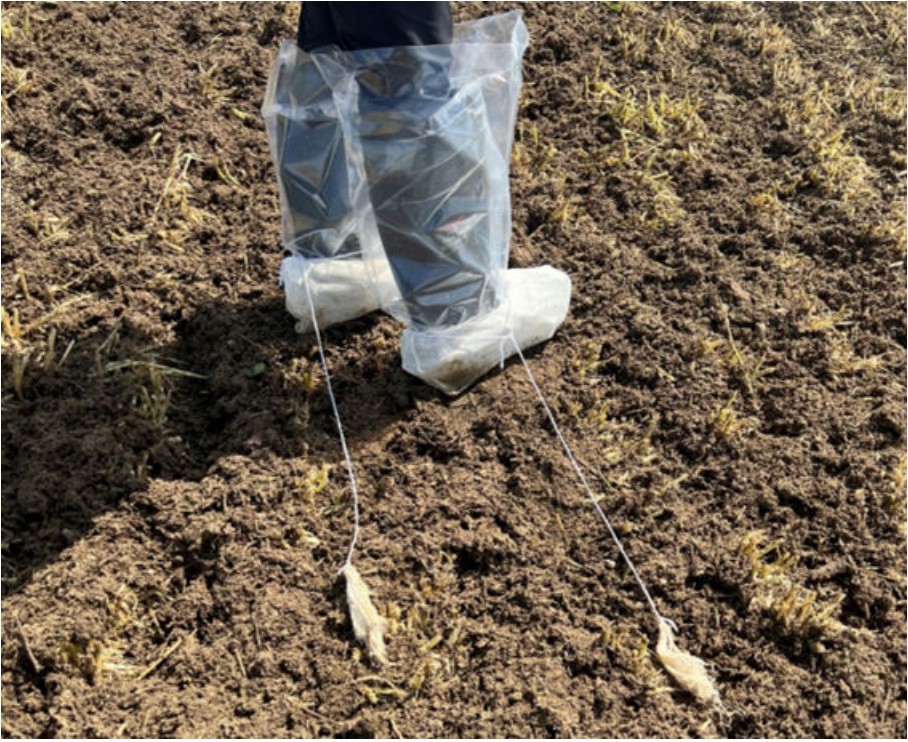

**FIG 5** Sample collection setup for one repetition. Left foot: one bootie hydrated with BPW and one drag swab hydrated with BPW. Right foot: one bootie hydrated with PBS and one drag swab hydrated with PBS. Samplers also wore clean plastic boot covers under each bootie swab. Not pictured but collected: composite soil grab using a small, disinfected, handheld metal spade and a sterile bag.

the sample collection path. Each repetition was a 300-m walk in a straight line. At the end of each 300-m walk, the sampler then used the adjacent empty bed to change direction and collect the next repetition of samples, walking back across the length of the field. The walking route creates a serpentine pattern across the field, walking down and back across the field. Samples were shipped to Illinois and processed within 36 h of sampling.

The next 18 repetitions of samples were collected in July 2024 across two sampling dates in a small-scale mixed horticultural field in central Illinois, measuring approximately 75 m long by 100 m wide. Samples collected on this field were divided into three different zones based on commodity type, where zone one was beets; zone two was mixed leafy greens (herbs, lettuce, and cabbage); and zone three was various types of sweet peppers grown above black paper. Each zone consisted of sample paths totaling approximately 75 m per repetition (Fig. S4).

The final 10 repetitions were collected over two sampling dates in an apple orchard in southern Illinois (five repetitions per day) during the harvest season of August 2024. The walking path covered approximately 100 m by walking under the canopy of 10 consecutive trees approximately 1.5 m from the trunk (Fig. S5).

## Sample processing, plating, and enumeration

Samples were preprocessed to obtain the initial dilution for enumerating indicator organisms, including TC, APC, and *E. coli*. For each bootie and drag swab sample, the mass of the dry swab was measured before sampling. The mass of soil collected by the swabs was calculated by subtracting the initial mass of the swab from the final mass of the complete sample after collection using the same method as in references 10 and 13. Soil mass collected by the drag swabs averaged 9.5 ± 3.8 g, and bootie swabs averaged 22.0 ± 15.0 g across all 48 repetitions. Within 18–36 h of collection, samples were then diluted by fixed volumes of 20 and 50 mL of sterile PBS to the drag and boot covers, respectively. Dilution range for drag swabs was from 1:2.0 to 1:4.6 dilution based on the actual collected soil mass. Dilution range for the boot swabs was from 1:1.8 to 1:26.2. Soil grabs were thoroughly mixed, and 25 g sub-samples were removed and diluted with 100 mL of PBS to create 1:5 dilutions for enumeration.

Samples were mixed by massaging by hand for approximately 2 min, and the dilutant was extracted from the Whirl-Pak bags containing the swabs and sample using a sterile micropipette. The first 1 mL of the initial dilutant was used for direct plating on CHROMagar (CHROMagar ECC, La Plaine Saint-Denis, France) and standard methods agar (Criterion, New York, NY). An additional 100 µL was used for serial dilutions in PBS, and 100 µL of relevant dilutions was spiral plated. Standard method agar plates were incubated at 35 ± 2°C for 48 h. CHROMagar plates were incubated at 30 ± 2°C for 24 h per manufacturer's recommendations to maximize total coliform detection. All plates were manually counted, as the CHROMagar required differentiation between the blue colonies (*E. coli*) and purple colonies (TC).

## Enrichment of samples for the detection of *E. coli*

Next, 2× EC broth (Sigma-Aldrich, St. Louis, MO) was added in equal volume to the remaining PBS in the sample bag containing the swab or soil sample, and the bag was incubated at 45°C for 24 h. After incubation, enriched sample bags were massaged by hand for 30 s. Samples were then streaked to isolation on CHROMagar and incubated at 37 ± 2°C for 24 h per manufacturer's recommendations for general streak plate incubation. Plates with any blue colonies, indicating the presence of *E. coli*, were counted as positive.

## Data analysis

Plate counts in CFU/g soil sample mass were $\log_{10}$ transformed as a standard microbiology practice since raw counts in CFU/g typically are more normally distributed once log transformed (31, 32). The limit of detection (LOD) was calculated for each sample

using a <1 CFU count threshold and the initial sample dilution factor for the mass of soil collected. After the log transformation, 1/2 LOD was used for statistical analysis, which is consistent with the EPA (33) and the approach in previous aggregate sampling work in beef trim (21).

All statistical analyses and graphs were made using JMP Pro 18 (SAS Institute Inc., 2024) or R version 4.3.3. Scatterplots were used to visualize whether there was a difference in each wetting agent's ability to detect APC and TC and a linear regression to calculate $R^2$ and Pearson's correlation coefficients. A matched-pair comparison $t$-test was run between paired sample types (bootie and drag swabs) by microbial test (TC and APC) to understand if the wetting agents (PBS and BPW) performed differently in each microbial test by sample type. A false discovery rate (FDR) correction was conducted to control for false positives (Type 1 error) across the data for a more reliable set of results (34). The same quantitative statistical analysis was not conducted on the *E. coli* count data, as most individual samples across all sampling trips were <LOD (74.5% of samples); therefore, the data are highly censored. Instead, a chi-square test was used to analyze enrichment results for *E. coli* collected by each collection method and wetting agent. A chi-square test is appropriate for analyzing 2 × 2 contingency tables that contain different sample sizes (35).

An analysis of variance (ANOVA) model compared the differences in detection of indicator organisms by sample collection method for each commodity. A Tukey's honest significant difference (HSD) was used to assess the significance of differences between the group means. An additional ANOVA was conducted on the overall differences between commodities for the mean difference of bootie swabs and soil grabs or drag swabs and soil grabs by indicator, and significant differences between groups are indicated by another Tukey's HSD. Paired $t$-tests were conducted to test if the mean differences between bootie swabs and soil grabs or drag swabs and soil grabs are statistically different from zero, and $P$-values adjusted for FDR correction.

## DNA extraction, amplification, sequencing, and analysis from soil samples

A total of 150 samples ($n = 30$ samples from each commodity type) were selected for DNA extraction. All bootie, drag, and soil grabs for the beets, peppers, and leafy greens were selected ($n = 90$ total, 30 for each commodity type). From the melon farm, $n = 30$ (10 booties, 10 drags, 10 soil grabs) out of 100 total samples were randomly selected using a random number generator. Similarly, a total of $n = 30$ (10 booties, 10 drags, 10 soil grabs) out of 50 total samples were also randomly selected from the apple orchard. Of the 150 samples where DNA was extracted, only 144 were sequenced, as six of the samples did not contain enough DNA for sequencing per the submission guidelines of the Roy J. Carver Sequencing Center. The six rejected samples all originated from the melon farm; however, they were evenly distributed across drag, bootie, and grab samples (two of each).

DNeasy PowerSoil Pro Kit (QIAGEN, Maryland) was used to extract the soil microbiota DNA following the manufacturer's directions. The purity and preliminary concentration of the extracted samples were measured by a NanoDrop 2000c spectrophotometer (Thermo Scientific, California); the base pair length was analyzed using gel imaging; and lastly, the concentration was measured using the Quant-iT PicoGreen dsDNA Reagent and Kit (Invitrogen, California). Extracted samples were diluted to 15–49 ng/µL and stored at −20°C before submitting to the Roy J. Carver Biotechnology Center DNA Services Laboratory at the University of Illinois at Urbana-Champaign. DNA amplifications were done via Fluidigm with 16S V3–V4 targets. Sequencing was conducted using Illumina MiSeq V2 (Illumina Inc., USA).

Microbial community analysis for alpha and beta diversities was performed by the High-Performance Computing in Biology (HPCBio) group at the Roy J. Carver Biotechnology Center at the University of Illinois. Data were imported and processed in R version 4.4.1, with quality control checks using DADA2 (36). Overall compositional summary for all unfiltered groups grouped by sample collection method and wetting agent can

be found in Fig. S2. Data were passed through filters to remove artifacts, prune out low-count taxa, and remove any possible bad or uninformative samples using phyloseq (version 1.48.0) package in R. No full samples were filtered out. A total of 26,854 taxa were available for further analysis after filtering. Alpha diversity metrics were observed and Chao1, accounting for richness both with and without rare taxa. Assumptions of the parametric test were tested and not met. Specifically, we used the Shapiro-Wilk test in R to test the normality of the data and Koenker's test to check for homoscedasticity. Therefore, nonparametric tests (Kruskal-Wallis and Wilcoxon rank sum tests) were used to evaluate differences in diversity metrics between the wetting agents and sample types. Principal coordinate analysis (PCoA) showing Bray-Curtis distance (measure of beta diversity) was created, and statistical significance was tested using permutational multivariate analysis of variance (PERMANOVA) (37) for wetting agent and sample type.

## Conclusion

This study shows that bootie and drag swabs hydrated in PBS or BPW are potentially a more representative and practical method of soil sampling and monitoring in comparison to conventional soil grabs. Aggregative soil sampling can sample large areas of produce fields in a non-destructive manner and is able to consistently detect higher levels of indicator organisms in a variety of ground conditions for different produce commodities. Detecting these indicator organisms could play a role in preharvest soil monitoring of the microbial communities of produce fields, which will enable growers to be aware of the potential presence of fecal contamination or pathogenic microorganisms in the soil to improve food safety plans.

## ACKNOWLEDGMENTS

Funding for this project was made possible by The Center for Produce Safety project 2024CPS08. Any opinions, findings, conclusions, or recommendations expressed in this publication (or audiovisual) are those of the author(s) and do not necessarily reflect the view of The Center for Produce Safety (https://www.centerforproducesafety.org/research-database/testing-wetting-agents-for-soil-drag-and-bootie-swabs-and-validating-them-in-varied-agricultural-soils).

We would like to thank Andrew Margenot, PhD, from the Department of Crop Sciences at the University of Illinois at Urbana-Champaign. We would also like to thank Christopher Fields, PhD, from the HPCBio Group from the Roy J. Carver Biotechnology Center at University of Illinois at Urbana-Champaign.

## AUTHOR AFFILIATIONS

[1]Department of Food Science and Human Nutrition, University of Illinois Urbana-Champaign, Urbana, Illinois, USA
[2]High Performance Computing in Biology, Roy J. Carver Biotechnology Center, University of Illinois Urbana-Champaign, Urbana, Illinois, USA

## AUTHOR ORCIDs

Erin Kealey http://orcid.org/0009-0000-4130-8448
Matthew J. Stasiewicz http://orcid.org/0000-0003-2712-0793

## FUNDING

| Funder | Grant(s) | Author(s) |
| --- | --- | --- |
| Center for Produce Safety | 2024CPS08 | Matthew J. Stasiewicz |

## AUTHOR CONTRIBUTIONS

Erin Kealey, Data curation, Formal analysis, Investigation, Methodology, Validation, Visualization, Writing – original draft, Writing – review and editing | Ray Elementi, Data curation, Formal analysis, Investigation, Methodology, Visualization | Zemme Frankowski, Investigation, Visualization | Negin Valizadegan, Data curation, Software, Writing – review and editing | Cecil Barnett-Neefs, Investigation, Software, Visualization | Jiaying Wu, Conceptualization, Data curation, Methodology, Supervision, Writing – review and editing | Pratik Banerjee, Resources | Matthew J. Stasiewicz, Conceptualization, Funding acquisition, Project administration, Resources, Writing – review and editing

## DATA AVAILABILITY

All raw data and analysis files are available as a Git repository at https://github.com/foodsafetylab/Kealey-2025-PreharvestProduceSoilSampling. In addition, sequencing data are available at the NCBI Sequence Read Archive under BioProject ID PRJNA1380470.

## ADDITIONAL FILES

The following material is available online.

### Supplemental Material

**Supplemental figures and tables (Spectrum01663-25-S0001.docx).** Figures S1 to S5 and Tables S1 and S2.

### Open Peer Review

**PEER REVIEW HISTORY (review-history.pdf).** An accounting of the reviewer comments and feedback.

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
