## [Reviewer comments · Microbiology Spectrum]

Microbiology Spectrum

Validating Aggregative Soil Sampling Using Bootie and Drag Swabs Hydrated with Simple Wetting Agents in Commercial Produce Fields

Erin Kealey, Ray Elementi, Zemme Frankowski, Negin Valizadegan, Cecil Barnett-Neefs, Pratik Banerjee, Jiaying Wu, and Matthew Stasiewicz

Corresponding Author(s): Matthew Stasiewicz, University of Illinois Urbana-Champaign

Review Timeline:

Submission Date:	May 29, 2025
Editorial Decision:	September 23, 2025
Revision Received:	November 18, 2025
Accepted:	December 8, 2025

Editor: Kalliopi Rantsiou

Reviewer(s): The reviewers have opted to remain anonymous.

Transaction Report:

DOI: <https://doi.org/10.1128/spectrum.01663-25>

Re: Spectrum01663-25 (Validating Aggregative Soil Sampling Using Bootie and Drag Swabs Hydrated with Simple Wetting Agents in Commercial Produce Fields)

Dear Dr. Matthew Jon Stasiewicz:

Thank you for the privilege of reviewing your work. Below you will find my comments, instructions from the Spectrum editorial office, and the reviewer comments.

The manuscript has been reviewed by an expert reviewer and you can find the comments below. Also, I have read it and in addition to the reviewer's comments please consider the following:

- refine the objectives: is it first a comparison between wetting agents and then a comparison of sample types? Or would it be more linear to first compare sample types (bootie, drag and soil) and then in more detail the wetting agent?
- based on the comment above, consider inverting the order of presenting the results; first by sample type, then by wetting agent (figure 4 first, then figures 2 and 3).
- in relation to the different commodities considered, in the discussion relevant considerations are made regarding the risk of contamination from the soil. Likely, also the various agronomic operations are very different based on the commodity and such operations (depending also on the timing they are performed) could influence the outcomes of the sampling. In addition, seasonal and climatic variability could influence the efficiency and outcome of the sampling (for example wet soil as opposed to dry soil). In the manuscript no reference is provided regarding the time of the year sampling took place. Could you please comment?
- in table 2 the pair wise comparison between bootie and drag sampling appears to be missing. Is this intentional? Can you please explain?
- only a subset of samples were subjected to DNA extraction and amplicon sequencing. Is there a reason for this?

Revision Guidelines

Sincerely,
Kalliopi Rantsiou
Editor
Microbiology Spectrum

Reviewer #1 (Comments for the Author):

The submitted manuscript by Kealey et al. presents the results of work comparing novel sample collection techniques and buffers with standard soil sampling protocols to understand pathogen burden in agricultural fields. They show that PBS and BPW do not result in noticeable differences in surrogate organisms detected, and that both booty and drag swab sampling methods outperform collecting soil samples. The results can inform ongoing sampling protocols and reduce burden for agricultural workers. I have some general and line-by-line comments listed below.

1. The introduction rapidly jumps into improving soil sampling techniques but does not first present existing sampling techniques and areas where it can be improved. Suggest starting with a discussion of existing protocols and the general burden of foodborne illness from contaminated foods to set the stage for why we need better sampling protocols.
2. Suggest moving the materials and methods before the results.
3. Generally, statistical analyses need to be better described. The category of test was often described without giving the exact test used and comparisons made. Additionally, numerous tests were run on the same comparisons, which creates the multiple comparisons problem. The authors need to 1) decide which hypotheses they want to test and 2) pick the test that fits their data and can test that hypothesis. How data were calculated needs to be better described in the text and figure captions.
4. Is a general limitation of this work that drag swabs were "behind" the booties? Soil would've been doubly sampled but the booties were always the first sample? Could change what is detected and the amount detected. Since they are both relatively low burden on the sampler, could suggest combining both into a single composite.
5. Figure captions need to stand alone and better describe how those results were obtained and what we are looking at. The colors also need to be checked for color-blind friendliness.
6. Line 63: suggest explaining here what a bootie swab is, not sure all readers will know what you're talking about
7. Results section: data need to be better presented here. Often results were stated without giving specific numbers or only in reference to looking at a figure. Much of it is written like a summary of the results rather than a presentation of the results of your study
8. Lines 88 - 91: p-values on your correlation tests? What test did you run for correlation analysis?
9. Line 97 - 98: great observation! Statistical significance does not always mean biological relevance
10. Lines 136 - 138: why the different number of samples for bootie and drag vs soil? Wouldn't this affect the results of a Fisher's exact test?
11. Line 287: did you also measure the weight of the wetted swabs? How much of the weight is due to soil and not soaking in PBS or BPW?
12. Lines 290 - 291: how did you elute off of the swabs?
13. Line 317: what test did you run for this? Wilcoxon?
14. Line 323: suggest changing "differences in recovery" to "differences in detection" because recovery implies detecting something that was seeded into (added to) the matrix before sampling.
15. Lines 336 - 337: how were these 6 samples without enough DNA distributed between the sample locations? Is there a certain location that produced less DNA?
16. Lines 354 - 355: why did you look at all alpha diversity metrics? Suggest picking the one that is best for 16s sequencing and justifying that choice and not looking at them all. See previous comment for multiple comparisons problem.
17. Figure 2: remove the $y = 1x$ line, it is confusing. Suggest making the shapes larger and adding some transparency so we can better see overlap. Can also use a shape with an outline and change the fill for better distinguishing. Currently I cannot tell which is a triangle or which is a circle very easily.
18. Figure 3 top panel: Sub header says Kruskal-Wallis test, but the p-value in the panels says Wilcoxon, which did you run? Again, pick one alpha diversity measure that is best for 16s.
19. Figure 3 & 4 bottom panels: what do the ellipses show? 68% coverage? 90% coverage?
20. Table 1: need to better explain this table with the caption and in the text. What is the FDR correction? How did you calculate differences?
21. Table 2: the letters are impossible to interpret, not sure what the results are for this table.

Editor's Comments:

- refine the objectives: is it first a comparison between wetting agents and then a comparison of sample types? Or would it be more linear to first compare sample types (bootie, drag and soil) and then in more detail the wetting agent?

Reply: The objectives have been updated to state the primary objective to evaluate the efficacy of the wetting agents for swabs, then the comparison of sample types. This change has been updated in the Introduction.

- based on the comment above, consider inverting the order of presenting the results; first by sample type, then by wetting agent (figure 4 first, then figures 2 and 3).

Reply: The ordering of objectives are first wetting agent, then sample type. Therefore, the ordering of the figures have remained the same based on the revised objective statement.

- in relation to the different commodities considered, in the discussion relevant considerations are made regarding the risk of contamination from the soil. Likely, also the various agronomic operations are very different based on the commodity and such operations (depending also on the timing they are performed) could influence the outcomes of the sampling. In addition, seasonal and climatic variability could influence the efficiency and outcome of the sampling (for example wet soil as opposed to dry soil). In the manuscript no reference is provided regarding the time of the year sampling took place. Could you please comment?

Reply: The melons were sampled in May 2024, the mixed commodity farm (leafy greens, beets, peppers) were sampled in July 2024, and the apples were sampled in August 2024. This additional context was added to the manuscript in the methods section for each respective commodity.

- in table 2 the pair wise comparison between bootie and drag sampling appears to be missing. Is this intentional? Can you please explain?

Reply: Table 2 has been re-formatted for clarity, and broken into new Table 2 and Table 3, with additional information moved to supplemental. The pairwise (booties – grabs and

drags-grabs) comparison columns of Table 2 are now isolated as the new table 3. The lower-case subscript letters in new Table 2 show the Tukey's HSD ranking of different sampling methods within a commodity type. The only instance where there is a significant difference between drags and booties is for apples. This observation is in the results section of the manuscript.

- only a subset of samples were subjected to DNA extraction and amplicon sequencing.

Is there a reason for this?

Reply: A subset was selected due to budgetary restrictions.

Reviewer #1

The submitted manuscript by Kealey et al. presents the results of work comparing novel sample collection techniques and buffers with standard soil sampling protocols to understand pathogen burden in agricultural fields. They show that PBS and BPW do not result in noticeable differences in surrogate organisms detected, and that both booty and drag swab sampling methods outperform collecting soil samples. The results can inform ongoing sampling protocols and reduce burden for agricultural workers. I have some general and line-by-line comments listed below.

Reply: Thank you for your review of this manuscript. The authors appreciate the feedback and will address all comments below.

1. The introduction rapidly jumps into improving soil sampling techniques but does not first present existing sampling techniques and areas where it can be improved. Suggest starting with a discussion of existing protocols and the general burden of foodborne illness from contaminated foods to set the stage for why we need better sampling protocols.

Reply: Additional context and citations related to soil sampling in the produce industry and the burden of foodborne illness from produce were added in the introduction.

2. Suggest moving the materials and methods before the results.

Reply: The directions for authors manual suggest having the results immediately following the introductory sections and having the materials and methods section after the discussion. The sections can be moved if required, at the recommendation of the Editor.

3. Generally, statistical analyses need to be better described. The category of test was often described without giving the exact test used and comparisons made. Additionally, numerous tests were run on the same comparisons, which creates the multiple comparisons problem. The authors need to 1) decide which hypotheses they want to

test and 2) pick the test that fits their data and can test that hypothesis. How data were calculated needs to be better described in the text and figure captions.

Reply: The authors revised the data analysis section of the methods, figures and table legends, and results section to better describe the analyses conducted. Please see tracked changes in the text throughout this manuscript.

4. Is a general limitation of this work that drag swabs were "behind" the booties? Soil would've been doubly sampled but the booties were always the first sample? Could change what is detected and the amount detected. Since they are both relatively low burden on the sampler, could suggest combining both into a single composite.

Reply: Thank you for this suggestion. This is a limitation of this study, and a line in the discussion section have been added to address this topic.

5. Figure captions need to stand alone and better describe how those results were obtained and what we are looking at. The colors also need to be checked for color-blind friendliness.

Reply: Figure captions have been updated to better describe what is being presented. The colors have also been updated for colorblind friendliness.

6. Line 63: suggest explaining here what a bootie swab is, not sure all readers will know what you're talking about

Reply: An additional sentence was added to provide a better description of a bootie swab in this section.

7. Results section: data need to be better presented here. Often results were stated without giving specific numbers or only in reference to looking at a figure. Much of it is written like a summary of the results rather than a presentation of the results of your study

Reply: This section has been updated to better discuss the specific, numeric results from this study, in addition to table and figure captions and legends.

8. Lines 88 - 91: p-values on your correlation tests? What test did you run for correlation analysis?

Reply: Figure 2 shows a linear regression. The authors wished to demonstrate the correlation between the swabs and booties that used the same wetting agent and describe the strength of the relationships using the R^2 and correlation coefficients. A citation justifying the use of Pearson's correlation coefficient has been added to specify a weak vs strong correlation.

9. Line 97 - 98: great observation! Statistical significance does not always mean biological relevance

Reply: Thank you for this comment.

10. Lines 136 - 138: why the different number of samples for bootie and drag vs soil? Wouldn't this affect the results of a Fisher's exact test?

Reply: For each repetition, there is 1 PBS Bootie, 1 PBS drag, 1 BPW bootie, 1 BPW drag, and 1 soil grab (paired). Therefore, as there are 4 different aggregative swab samples per 1 soil grab ($4 \times 4 = 16$). Each repetition collected is described in the "Sample Collection Method" section in Materials and Methods. In addition, while reviewing this comment, the authors realized a more appropriate test would be a Chi-square test, which is still appropriate for 2×2 contingency tables with different sample sizes, but is better for describing larger sample sizes > 5 . This has been revised in the text and corresponding figure. The outcome is still the same where the soil grabs are significantly different than the aggregative swabs, and the p-value has only slightly changed.

11. Line 287: did you also measure the weight of the wetted swabs? How much of the weight is due to soil and not soaking in PBS or BPW?

Reply: The authors have rephrased this section for clarity. The methodology of the sample processing was kept consistent with Wu et al 2023 and 2025, and the citation was added.

12. Lines 290 - 291: how did you elute off of the swabs?

Reply: The swabs were eluted by using a micropipette (with sterilized tips) to pull the respective volume from the whirlpak bag containing the used swab and added dilutant. This detail was added to the "Sample Processing, Plating, and Enumeration" section of Materials and Methods.

13. Line 317: what test did you run for this? Wilcoxon?

Reply: A matched pairs comparison, a type of t-test, was conducted on the mean difference of APC and TC detected for each wetting agent. The mention of the t-test was added to this section.

14. Line 323: suggest changing "differences in recovery" to "differences in detection" because recovery implies detecting something that was seeded into (added to) the matrix before sampling.

Reply: Thank you for this suggestion. This change has been made in the text.

15. Lines 336 - 337: how were these 6 samples without enough DNA distributed between the sample locations? Is there a certain location that produced less DNA?

Reply: The 6 rejected samples were at the melon farm, however the rejected were equal across drags, booties, and grabs (2 of each). This additional information was added.

16. Lines 354 - 355: why did you look at all alpha diversity metrics? Suggest picking the one that is best for 16s sequencing and justifying that choice and not looking at them all. See previous comment for multiple comparisons problem.

Reply: Thank you for this observation. As the authors are more focused on the microbial richness that was detected in these samples, we have selected to focus on the Observed and Chao1 alpha diversity metrics. This has been further elaborated in the data analysis section, results, and in the corresponding figures.

17. Figure 2: remove the $y = 1x$ line, it is confusing. Suggest making the shapes larger and adding some transparency so we can better see overlap. Can also use a shape with an outline and change the fill for better distinguishing. Currently I cannot tell which is a triangle or which is a circle very easily.

Reply: This line has been removed for additional clarity. The colors of the shapes have been changed to be color-blind friendly.

18. Figure 3 top panel: Sub header says Kruskal-Wallis test, but the p-value in the panels says Wilcoxon, which did you run? Again, pick one alpha diversity measure that is best for 16s.

Reply: The authors removed the Kruskal-wallis line on the figure. Only a Wilcoxon was run, and that text was left over from a previous revision. The figure should now only contain Wilcoxon.

19. Figure 3 & 4 bottom panels: what do the ellipses show? 68% coverage? 90% coverage?

Reply: The ellipses show 95% confidence interval based on the `stat_ellipse()` from the `ggplot` package in R. The figure legend has been updated to state the 95% confidence interval. The figure legends have been updated to reflect this information.

20. Table 1: need to better explain this table with the caption and in the text. What is the FDR correction? How did you calculate differences?

Reply: Additional information has been added to Table 1 to better explain the comparisons being made as well as FDR correction. In addition, FDR correction is done on statistical tests in order to account for potential false-positives (Type 1 errors). Further elaboration was added to text with a citation.

21. Table 2: the letters are impossible to interpret, not sure what the results are for this table.

Reply: Table 2 has been reformatted for additional clarity, where it is now the new Table 2 and Table 3, with additional statistics moved to supplementals. The columns of new Table 2, which shows the Booties – Grab and Drags – Grab differences, along with their corresponding ANOVA and paired t-tests. The new table 3 has the lowercase letters only to compare the amount of indicators collected for the sample collection methods for each commodity. The footnote and caption have been updated for additional clarity. Additional analysis was added to the results section.

Re: Spectrum01663-25R1 (Validating Aggregative Soil Sampling Using Bootie and Drag Swabs Hydrated with Simple Wetting Agents in Commercial Produce Fields)

Dear Dr. Matthew Jon Stasiewicz:

No information is provided regarding availability of the sequencing data. Please provide such information prior to publication. Raw sequencing data should be uploaded in a public database.

Your manuscript has been accepted, and I am forwarding it to the ASM production staff for publication. Your paper will first be checked to make sure all elements meet the technical requirements. ASM staff will contact you if anything needs to be revised before copyediting and production can begin. Otherwise, you will be notified when your proofs are ready to be viewed.

Sincerely,
Kalliopi Rantsiou
Editor
Microbiology Spectrum